# Deep Feature Engineering in Colposcopy Image Recognition: A Comparative Study

**DOI:** 10.3390/bioengineering10010105

**Published:** 2023-01-12

**Authors:** Shefa Tawalbeh, Hiam Alquran, Mohammed Alsalatie

**Affiliations:** 1Department of Biomedical Systems and Informatics Engineering, Yarmouk University, Irbid 21163, Jordan; 2The Institute of Biomedical Technology, King Hussein Medical Center, Royal Jordanian Medical Service, Amman 11855, Jordan

**Keywords:** cervical cancer, feature fusion, feature selection, deep learning structures, support vector machine, disease discrimination accuracy, performance comparisons

## Abstract

Feature fusion techniques have been proposed and tested for many medical applications to improve diagnostic and classification problems. Specifically, cervical cancer classification can be improved by using such techniques. Feature fusion combines information from different datasets into a single dataset. This dataset contains superior discriminant power that can improve classification accuracy. In this paper, we conduct comparisons among six selected feature fusion techniques to provide the best possible classification accuracy of cervical cancer. The considered techniques are canonical correlation analysis, discriminant correlation analysis, least absolute shrinkage and selection operator, independent component analysis, principal component analysis, and concatenation. We generate ten feature datasets that come from the transfer learning of the most popular pre-trained deep learning models: Alex net, Resnet 18, Resnet 50, Resnet 10, Mobilenet, Shufflenet, Xception, Nasnet, Darknet 19, and VGG Net 16. The main contribution of this paper is to combine these models and then apply them to the six feature fusion techniques to discriminate various classes of cervical cancer. The obtained results are then fed into a support vector machine model to classify four cervical cancer classes (i.e., Negative, HISL, LSIL, and SCC). It has been found that the considered six techniques demand relatively comparable computational complexity when they are run on the same machine. However, the canonical correlation analysis has provided the best performance in classification accuracy among the six considered techniques, at 99.7%. The second-best methods were the independent component analysis, least absolute shrinkage and the selection operator, which were found to have a 98.3% accuracy. On the other hand, the worst-performing technique was the principal component analysis technique, which offered 90% accuracy. Our developed approach of analysis can be applied to other medical diagnosis classification problems, which may demand the reduction of feature dimensions as well as a further enhancement of classification performance.

## 1. Introduction

In 2020, 604,000 new cases of cervical cancer were estimated, and 342,000 deaths were reported; 90% of the new cases and deaths were reported in middle- and low-income countries [1]. These cases were due to the lack of health awareness as well as the limited access to screening methodologies. According to the World Health Organization (WHO), appropriate screening reduces morbidity and mortality among women [2]. In this regard, a pap smear is the most common early screening and diagnostic tool for cervical cancer. Hundreds of sub-pap smear images are examined under a microscope by a cytopathologist. This makes such manual analysis a subjective, error-prone, and time-consuming process.

Computer-aided design (CAD) tools can play an important role in overcoming the inconsistency, inaccuracy, and time-consuming problems of manual analysis. In the last few decades, automated methods have been developed and then approved by the food and drug administration (FDA) to diagnose and classify cervical cancer [3,4,5,6].

The recent advances in computing and the large growing data repository have supported efficient machine learning (ML) and deep learning (DL) algorithms to aid medical decisions. In recent years, pap smear images have been efficiently processed by adequate machine learning algorithms for cervical cancer classification [7,8,9,10,11,12]. One of the first steps in building such models is to identify the features that best describe the input data. 

In this paper, we mainly focus on providing comprehensive testing results for the estimation accuracy of various data fusion techniques when they are applied to cervical cancer classification. It is noted that data fusion can occur at different levels, such as the feature level, matching score level, or the decision level [13]. The main aim of feature fusion is to combine information from two or more feature sets into a single dataset that has more discriminant power than each feature vector. Accordingly, in this paper, we are interested in utilizing this discriminant power in separating classes more efficiently. We are conducting a comparative analysis to test the effectiveness of selected feature fusion techniques in enhancing the accuracy of cervical cancer classification. These techniques are applied on the feature level, which reduces the dimensionality of the feature datasets while enhancing the accuracy of classification.

The following literature review highlights recent studies that show the effectiveness of data fusion techniques for cervical cancer detection. However, due to the limited number of studies that use feature-level fusion for cervical cancer classification, which is the main purpose of this paper, the literature review is followed by other related studies that use feature fusion on other medical images for diagnostic and classification purposes.

## 2. Related Work

In this section, we have selected the most recent studies that use feature engineering specifically on cervical cancer classification. In each article, the authors used a different fusion technique and showed how this improved the classification accuracy. Alquran et al. [14], proposed a computer-aided diagnosis of cervical cancer classification based on feature fusion between the well-known Shuffle Net DL structure and a novel Cervical Net structure. The novel Cervical Net structure was proposed by Alquran. The authors used a principal component analysis (PCA) and canonical correlation analysis (CCA) as the feature reduction and fusion techniques. The resultant features were fed into different ML classifiers. The best accuracy of 99.1% was obtained using a support vector machine (SVM) to classify between five classes of pap smear images. On the other hand, Liu et al. [15] proposed a framework to classify cervical cancer cell classification based on DL. Specifically, they extracted local and global features using a convolutional neural network (CNN) module and a visual transformer module, respectively, from cervical cancer cell images. Then these features were fused using a multilayer perceptron module. The framework proposed by Liu et al. obtained an accuracy of classification of 91.72% by combining two datasets (CRIC and SIPaKMeD datasets) for an 11-class classification problem.

Rahman et al. [16] proposed a method for enhancing computer-aided diagnosis of cervical pap smear images using a hybrid deep feature fusion (HDFF) method. This method was tested on the SIPaKMeD dataset and performance was compared with multiple DL models alongside the late fusion method. The late fusion, sometimes called decision-level fusion, leverages predictions from multiple models to make a final decision. In their paper using the SIPaKMeD dataset, they obtained a classification accuracy of 99.85%, 99.38%, and 99.14%, for a 2-class, 3-class, and 5-class classification. They also tested their model on the Herlev dataset and achieved an accuracy of 98.32% for a 2-class and 90.32% for a 7-class classification. Moreover, Hussain et al. [17] proposed a computer-assisted screening system based on DL. The paper explored six deep learning structures, namely Alexnet, Vggnet (vgg-16 and vgg-19), Resnet (resnet-50 and resnet-101), and Googlenet architectures, for a four-class diagnosis of cervical cancer lesions. The authors fused the best three DL models yielding the best accuracy for class classification. The output of each deep learning structure mentioned above was evaluated based on performance, then the best three models (Resnet-50, Resnet-101, and Googlenet) were combined (fused) to generate their ensemble classifier. Their results showed that the proposed classifier achieved the highest area under curve (AUC) = 97% between two positive and negative classes. 

The above articles applied some sort of data fusion method to enhance the decision accuracy from cervical cancer pap smear images. However, not all the above studies used feature-level fusion. Rahman et al. and Hussain et al. used decision-level fusion. Alquran et al. used CCA to fuse features from two datasets, and finally, Lui et al. used a multilayer perceptron model. Due to the limited number of studies that use feature fusion for cervical cancer classification, we listed other studies that highlight the effectiveness of using feature fusion and reduction analysis to improve other medical image classification problems. In the below references, we have selected articles that used feature analysis CCA, discriminant correlation analysis (DCA), least absolute shrinkage and selection operator (LASSO), independent component analysis (ICA), PCA, and others. Most of the feature fusion techniques mentioned in the below articles were selected in our comparative study.

Zhang et al. [18] studied four different feature fusion and reduction techniques between two independent feature sets, namely, LungTrans features and PrRadomics features. In their paper, the authors proposed a method for feature fusion named the ‘risk score based’ feature fusion method. Their paper showed that the proposed risk score-based feature fusion method improves the prognosis performance for predicting the survival of pancreatic ductal adenocarcinoma patients, yielding an increase of 40% of AUC compared with AUC without fusion. The feature fusion and reduction techniques used were PCA, LASSO, Boruto, Univariant Cox proportional-hazards CPH, and the proposed risk score-based technique. The latest was performed by feeding each feature set to two different random forest classification models, and the resulting most significant features were fed into another random forest-based prognosis model. In summary, Zhang et al. compared five different feature fusion techniques on two feature datasets (lungtrans features and PrRadomics) to improve the prognosis of PDAC. Moreover, Fan et al. [19] integrated dynamic contrast-enhanced magnetic resonance imaging and T2-weighted imaging radiomic features by a CCA. The paper aimed to provide related complementary information between the fused feature datasets to improve breast cancer prediction. After fusing the two datasets, they used SVM-based recursive feature elimination (SVM-RFE) to identify the optimal features for prediction. They noticed an enhancement in the AUC after using fused features. Moreover, they reported that using CCA was more beneficial than using concatenation-based feature fusion or classifier fusion methods. Another method for feature-level fusion is the DCA, which was proposed by Haghighat et al. [20] where they introduce DCA as an effective feature fusion method to enhance class separation. They tested DCA on multiple biometric datasets showing the effectiveness of this approach. Using DCA combines the information from more than one feature dataset into a single dataset that has more discriminant power. This was applied to different medical diagnostic applications, for example, Wang et al. [21] extracted features from four datasets for COVID-19 CCT images using a novel feature learning algorithm. Then, they proposed a selection algorithm to select the best two models. Finally, they used the DCA to fuse the two features from the two models. The final determined model was named CCSHNET. Their proposed CCSHNET model based on fusing features using DCA showed high-performance measures when compared to other COVID-19 detection methods.

In this paper, we focus on the existing feature engineering techniques. The utilization of pre-trained DL structures to extract features from whole-slice pap smear images is a promising idea, alongside exploiting feature fusion and reduction techniques to obtain the highest level of confidential computer-aided diagnosis system for colposcopy images. To our knowledge, this is the first paper that employs ten deep-learning models to extract representative descriptors, which can be utilized for the recognition of pap smear image diseases via feature engineering algorithms. The novelty in our approach is using existing feature-level fusion to extract the most representative features from ten DL models to enhance classification accuracy.

## 3. Materials and Methods

The method that is proposed in this paper is illustrated in Figure 1.

The methodology followed in this paper consists of six steps. Step one: collect the cytology dataset that consists of 1000 samples for 4 different cervical cancer classes. Step two: perform image augmentation. Step three: extract features using CNN from ten deep learning structures (4 features for each DL structure total of 40 features). Step four: concatenate all the features from the ten DL structures to be fed into the feature fusion step. Step five: apply different feature fusion techniques to fuse or select features. Step six: feed the features into an SVM to measure classification performance. The details of each step are described in detail in the following section. Cytology dataset acquisition and augmentation are described in Section 3.1 and Section 3.2. Extracting features using deep learning structures are described in Section 3.3. The theoretical background of the six selected fusion techniques is described in Section 3.4 and Section 3.5. Finally, the SVM method is described in Section 3.6.

### 3.1. Image Acquisition

One of the cervical screening tests is liquid-based cytology (LBC). A total of 963 LBC images are separated into four sets to reflect the four classes, namely, NILM, LSIL, HSIL, and SCC, that make up the whole repository. It includes cervical cancer-related precancerous and cancerous lesions that meet the Bethesda System requirements (TBS). A total of 460 patients visited the obstetrics and gynecology (O&G) department of the public hospital with varied gynecological issues and were examined using the ICC50 HD microscope to take the images at a magnification of 40×. The pathology department’s professionals then examined and categorized the images [22].

### 3.2. New Image Augmentation

Data augmentation is a strategy used to expand the amount of data by adding slightly changed copies of either existing data or freshly created synthetic data from existing data. It serves as a regularizer and helps minimize overfitting. This paper used rotation images at random angles in the range of [−45, 45] degrees, image resizing with random scale factors between [0.2, 1], and translation in both directions X and Y are [−3, 3], to accomplish image augmentation for the abnormal cases [23]. Table 1 describes the number of images before and after augmentation.

### 3.3. Deep Learning Features

Several pre-trained deep learning models are employed to extract the most representative features from the last fully connected model in each one. The selected deep-learning structures were trained on the ImageNet database to distinguish between 1000 classes from nature. Transfer learning techniques were used to make these structures compatible with the designed problem statement, which focused on classifying four types of whole-slice cervical cells. The transfer learning appeared by augmenting the input size of the image to be appropriated with the input layer of each one and removing the last fully connected layer to make it four neurons for four classes. The represented features for each model are extracted from the last fully connected layer. Each one provides four distinguished features for four classes. The networks that are utilized for feature extractions are AlexNet, ResNet18,50, and 101, Mobile Net, Shuffle Net, Xception Net, Nasnet, Dark-19, and VGG16.

#### 3.3.1. AlexNet

AlexNet is one of the most popular convolutional networks. It was first introduced in 2012 for ImageNet recognition of 1000 nature classes. AlexNet architecture consists of five convolutional layers, three max-pooling layers, two normalization layers, and two fully connected layers with a softmax layer beside input and output layers. Each convolutional layer is composed of convolutional filters, which are responsible for extracting the graphical features, and a nonlinear activation function named ReLU. Max pooling is in charge of the down sampling of activated extracted features. The image input size should be 227 × 227 × 3 to accommodate the parameters of the following layers [24].

#### 3.3.2. ResNets

Residual neural networks (18, 50, and 101) are pre-trained convolutional neural networks. They are distinguished by their residual block property. This feature solves the problems of vanishing or exploding gradients due to deep learning. ResNets allow the formation of a skip connection, which enables the activation of a layer to further layers by skipping some layers in between. That is the architecture of the residual block. ResNets consist of stacking such blocks. Several versions of ResNet have existed that depend mainly on the number of connected layers, such as ResNet 18, ResNet50, and ResNet101. The input size of these networks is 224 × 224 × 3 [25].

#### 3.3.3. Mobile Net

Mobile Net is a pre-trained convolutional neural network. It was designed for mobile and computer vision applications. One of the most prominent properties is depth-wise separable convolution, which reduces the number of parameters that contain problems in the existing convolutional layers in the existing networks. That depends mainly on depth-wise convolution, which is named channel-wise spatial convolution, followed by pointwise convolution, with a kernel size of 1 × 1 that combines the resultant features from the depth-wise convolution. On the other hand, it reduces the dimension of generated feature map. Their advantages are low latency and a low number of parameters [26,27].

#### 3.3.4. Shuffle Net

Shuffle Net is one of the most efficient networks that is designed for mobile applications. To maintain a high level of accuracy, Shuffle Net performs point-wise group convolution and channel convolution. These distinguished properties make Shuffle Net more accurate, while reducing the complex time computation. It consists of a stacking of shuffle netblocks, each one consisting of two grouped convolutional layers, channel shuffle layer, in addition to depth-wise convolutional layers. The process within one block considers depth-wise convolutional and point-wise convolution as well. The output from each block passes to the ReLU layer for mapping purposes. The designed input layer is compatible with image size 224 × 224 × 3 [28].

#### 3.3.5. Xception Net

The insight behind the 3D convolutional layer is the capability to allow the filter to learn within the 2D spatial domain alongside the depth via channel dimension. Therefore, the output is obtained by the correlation between the spatial and the channel convolutions. The idea behind the inception blocks makes the process easy and forward by using several explicit series of operations ended by cross-channel correlation and spatial correlations. The process operation starts with cross-channel correlation to reduce the dimension via 1 × 1 convolution that maps the input data into 3 or 4 spaces that are lower dimensional than the original input space. After that, the process proceeds via regular 3 × 3 or 5 × 5 convolutions.

The new version of the inception module is called the “extreme”. The Xception module performs the channel convolution and obtains a spatial convolution for each channel separately. The Xception architecture consists of 36 convolutional layers forming the feature extraction base of the network. Moreover, the Xception structure is formed as linear stacking of inception modules [29].

#### 3.3.6. NasNet

Neural search architecture (NAS) networks stand for NASNET. It is a predefined architecture that is trained over an ImageNet database of over 1000 categories from nature. It consists of a series of cells. These cells are the normal and reduction cell, where the normal cell is responsible for constructing the feature map via convolutional filters, and the reduction cell oversees the reduction of the size of the feature map in terms of width and height by factor two. Moreover, the structure of NASNET ended by the softmax layer yields the probability for the last classification layer [30].

#### 3.3.7. Dark-19 Net

Darknet is one of the most known deep learning structures that is used to detect objects from images in the available dataset. Dark Net-19 consists of 19 layers, which yields to its name. The Darknet has various applications in object detection, alongside counting as the most known algorithm in YOLO, which stands for you only look once [31].

#### 3.3.8. VGG-16 Net

VGG stands for visual geometry group convolutional network, which is trained on the ImageNet database. VGG16 consists of 16 layers: thirteen are convolutional layers, and the rest are fully connected layers. The input layer is compatible in design with image size 224 × 224 × 3. The VGG network has a small perspective field where the convolutional filter size is 3 × 3, which influences capturing more details in the image in both left-right and up-down directions. Moreover, the convolution of 1 × 1 acts as a linear transformation for the input data. This network utilizes transfer learning techniques to extract the most significant features for four pap smear image classes [25].

### 3.4. Feature Fusion

Feature extraction is the genesis of the recognition between various classes in machine learning algorithms. However, the leverage of most representative features may appear in the performance of the designed classifier. Therefore, looking for the most influential attributes is a crucial challenge in computer-aided diagnosis systems. This paper compares techniques in engineering features to classify whole-slice images with highly confidential results. Employing deep learning semantic descriptors alongside one of the most known feature processing methods is a hot topic presented in this paper. This paper applies two types of fusion algorithms: CCA and DCA.

#### 3.4.1. Canonical Correlation Analysis

CCA is one state-of-art statistical analysis of multivariate data that measures the linear relationship between two datasets. It is one of the most commonly used methods in data fusion. CCA focuses on maximizing the correlation between the variables of the two datasets and ignores the relationship between the variables within the same datasets [32]. 

CCA is defined as two sets of basis feature vectors, where x and y, the correlation of the features between these bases, are mutually maximized.

These two datasets x and y can be written as linear combinations of their internal features: $$x=x^{T}\hat{w_{x}}\;y=y^{T}\hat{w_{y}}$$

To maximize the above two functions, the corresponding function should be maximized
$$\rho =\frac{E\left[\mathrm{xy}\right]}{\sqrt{E\left[x^{2}]\;E[y^{2}\right]}}$$

The maximum values of ρ in respect to the weights of subsets x and y are called canonical variates. 

#### 3.4.2. Discriminant Correlation Analysis

Feature fusion aims to find the highly correlated features between two separate datasets. In DCA, the class is considered a membership of correlation analysis that enhances the fusion process. DCA needs low computational complexity, which leads to minimizing time in real-world applications. Moreover, it reduces the number of features that best describes the original ones [21]. The corresponding equations illustrate the process of DCA. The training features are:$$E=\left\{\left(x_{1},y_{1}\right),\left(x_{2},y_{2}\right),\ldots ,\left(x_{n},y_{n}\right)\right\}\;,\;where\;y_{i}=\left\{1,\ldots ,k\right\}$$
where k is the number of classes, *x*, and *y* are features and their corresponding class.

The first step is calculating the mean of each class separately:$$\bar{x_{i}}=\frac{1}{m_{i}}\overset{m_{i}}{\underset{j=1}{\sum }}x_{j}^{i},$$
where $m_{i}$ is the number of samples in each class. Then, evaluate the overall mean of the training set by:$$\bar{x}=\frac{1}{n}\overset{k}{\underset{v=1}{\sum }}m_{v}\bar{x_{v}}$$

The covariance matrix is calculated by the following equation:$$sigma=\varphi^{T}\varphi ,$$
where $\varphi =\sqrt{m_{1}}\left(\bar{x_{1}}-\bar{x}\right),\ldots ,\sqrt{m_{k}}\left(\bar{x_{k}}-\bar{x}\right)$.

The singular value matrices (SVD):$$sigma=U\Lambda U^{T},$$
where Λ is the diagonal of eigenvalues $\lambda_{i}$, U donates to eigenvectors. The eigenvectors are ordered in ascending form with their corresponding eigenvectors. The transformation matrix is given by: $$R=\varphi U_{t}\Lambda_{t\times t}^{-1/2}$$

All previous steps are performed on each set separately. Then, data points are transformed as:$$Z_{1}=R_{1}^{T}X_{1}\;Z_{2}=R_{2}^{T}X_{2}$$

After that, the covariance matrix of transformed features between two sets:$$S_{b}=Z_{1}Z_{2}^{T}$$

Then, SVD is calculated for $S_{b}$:$$S_{b}=V\Sigma V^{T}$$

Then, the transformation matrix is given by:$$T=V\Sigma^{-\frac{1}{2}}$$

Then, the data are generated by the following equation:$$X_{1}^{\prime }=T^{T}Z_{1}\;X_{2}^{\prime }=T^{T}Z_{2}$$

Finally, the output features are generated according to the equations:$$X^{\prime }=X_{1}^{\prime }+X_{2}^{\prime }$$

### 3.5. Feature Selection

Feature selection is one of the prominent topics in machine learning, processing, and data analysis. The mean goal of attributes selection maintains that the best representative attributes have high variance, which reduces the dimensionality of the feature maps and reduces the time computation and complexity. Various feature selection techniques are proposed in the literature. In this paper LASSO, PCA, and ICA are used. Below is a description of each method.

#### 3.5.1. Least Absolute Shrinkage and Selection Operator

LASSO is one type of penalized logistic regression, where a penalty is imposed on the logistic model for having too many variables. This leads to shrinking the coefficients of the least contributive variables to zero. Specifically, LASSO forces the less contributive variables to become exactly zero. For LASSO regression, a constant lambda should be specified to adjust the amount of the coefficient shrinkage. The best lambda can be defined as the lambda that minimizes the cross-validation mean square error rate. The mean squared error (MSE) measures how close a regression line is to a set of data points. In our method, we have chosen the one standard deviation lambda λ1se to select the final model [33].

#### 3.5.2. Principal Component Analysis

The PCA is well known as an unsupervised learning algorithm used to obtain the most significant features using dimensionality reduction. First, the dataset is standardized using the Z score a

$z_{i}=\frac{x_{i}-\mu_{c}}{\sigma_{c}},$ where $x_{i}$ is the feature value for each sample, $\mu_{c}$ the mean of each feature column, and $\sigma_{c}$ is the standard deviation for each column as well.

Then, the covariance matrix is built for all standardized features, where the diagonal represents the variance of each feature, and the off-diagonal describes the covariance between two features in the whole dataset. Then, calculate the eigenvector and eigenvalues that represent the 95% variation for the constructing covariance matrix. Finally, the eigenvalues are ascended from the highest to lowest principal components. The projection is calculated to find the original significant features from the original dataset [34,35].

#### 3.5.3. Independent Component Analysis

ICA is a statistical technique that reveals hidden factors (sources) from sets of random variables, or signals [36], and these sources are maximally independent. ICA has been used in unsupervised learning classification problems. Many studies have shown the utility of ICA to extract independent features from the original feature dataset to reduce the feature space and thus, improve classification accuracy [37,38,39]. Mathematically speaking, assuming that x(t) = x1(t); x2(t); …; xn(t) are the set of observed variables that are a combination of the original and mutually independent sources (original features), source s(t) = s1(t); s2(t); …; sn(t), the relation can be expressed by x(t) = As(t), where A is called the mixing matrix. In other words, the equation can be written as y = Wx, where W is the demixing matrix W = A^−1^, and y = y1; y2; …; yn, are the independent components. The demixing matrix and the independent variables can be found from mixed observations using one of the ICA algorithms such as fastICA [40], which was used in this paper. Furthermore, the set of extracted components (y = y1; y2; …; yn) are non-Gaussian and maximally independent. One way to measure this is using the kurtosis [41] measure, which was adopted in this paper to rank the extracted independent components.

### 3.6. Support Vector Machine Classifier

SVM classifier is a well-known supervised machine learning algorithm, which was developed in 1963 by Vladimir N. Vapnik. SVM selects the extreme training points from different classes to specify the boundary region between various labels, which is called the margin region. If the training points are linearly separable, then the discrimination between them is an easy task. If it is not linearly separable, then the SVM has a distinguished property to represent this feature into higher space using the kernel trick to be linearly separable in higher space. These kernels are radial basis functions, polynomial-Gaussian, and many forms of kernels.

## 4. Results & Discussion

The whole-slice images are passed independently to ten pre-trained deep learning structures. Each pre-trained CNN is modified using transfer learning so that the last fully connected layers become compatible with four classes. Four features were extracted from each CNN. The generated feature map consists of 40 features from 1000 samples. Each class consists of 250 samples; 250 slices of the normal class; 250 samples for HISL; samples for 250 LSIL; and samples for 250 SCC. 

The generated maps are passed to different feature selection and fusion methods. The resultant feature map is divided into a 70% training set for the SVM classifier and a 30% to test the generated SVM model. The corresponding results describe the performance of the SVM in discriminating four colposcopy whole-slice images using feature fusion and selection techniques.

### 4.1. CCA

The whole mapped features were passed to the CCA, which resulted in the six most correlated attributes. These were then split into 70% as a training set and 30% as a test set for the SVM classifier. The resultant confusion matrix shown in Figure 2a shows the performance of the trained model. The HSIL samples are classified correctly with a sensitivity of 100% and a precision of 98.7%. Moreover, the LSIL achieves 100% positive predictive value (PPV) and 100% recall. The same prominent results are obtained in the normal class, with a true positive rate of 100% and precision of 100%. For the lowest sensitivity obtained in the SCC, the PPV is 100%. Finally, the overall accuracy achieved is 99.7. Figure 3 illustrates the receiver operating characteristics (ROC), which defines the area under the curve (AUC) for each feature selection technique. The AUC represents the relation between the false positive rate (specificity) on the x-axis and the true positive rate (sensitivity or recall) on the y-axis for each class. As is clear from Figure 3a, the AUC for all classes in the case of the CCA is one. 

### 4.2. DCA

The same procedure was performed for the discriminant correlated analysis. Forty DL-labeled features were passed to the DCA. The performance of the trained SVM model reached 98.7% for sensitivity to the HISL category, with a low PPV of 96.1%. However, the prominent results appear in both the LSIL and normal classes, where recall and precision reach 100%. The behavior of the designed classifier in the SCC samples is similar to the HSIL, with the lowest sensitivity of no more than 96%, and a precision of 98.6%. The overall accuracy of the SVM using the DCA feature fusion method is 98.7%. The confusion matrix is shown in Figure 2b. On the other hand, the performance of the combination between the DCA and SVM is represented in Figure 3b. Almost all classes have the highest level of AUC.

### 4.3. LASSO

The feature set was passed to the LASSO algorithm to select the most representative features. Figure 4 shows the cross-validated mean square error (MSE) for the LASSO model. Each red dot represents a lambda (λ) value with confidence intervals for the error rate. Two vertical lines are drawn between the lambda that achieves the lowest MSE (λmin) and the lambda that indicates the highest value of MSE within one standard deviation of the minimum lambda (λ1se). The numbers at the top of the plot represent the number of features of the model at a given lambda value.

In our methodology, we have selected λ1se = 0.004 to be fed into the LASSO model, which resulted in the extraction of 19 features from a total of 40. Therefore, the selected features that passed to the SVM were 19. The corresponding confusion matrix is shown in Figure 2c, which clarifies the performance of the SVM model using the 19 selected features by the LASSO algorithm. The SVM correctly distinguishes LSIL, with higher sensitivity and precision reaching 100%. However, the lowest true positive rate in the HISL class and its PPV do not exceed 97.3%. The performance of the normal class is 98.7% and 100% for recall and precision, respectively. Furthermore, the SCC has the lowest precision of 96.1% and a moderate value sensitivity of 97.3%. Moreover, the AUC for all classes is almost equal to one. This shows the effectiveness of the proposed method. 

### 4.4. Feature Concatenation

The feature concatenation is performed by unionizing all features into a single dataset. All deep learning features are concatenated to obtain 40 attributes, which are split into 70% for SVM training and the rest to evaluate the classifier. The corresponding confusion matrix shown in Figure 2d illustrates the outputs of the test data using the fused 40 features. It is clear from the confusion matrix of the fused 40 features, that 72 cases of HSIL are classified correctly among the 75 cases, with recall reaching 96% and precision reaching 94.7%. For the LSIL 75 samples, they are classified correctly with a sensitivity and precision of 100%. The same applies for the normal classes, where the performance is 100% in both the TPR and PPV. The worst behavior appeared in the SCC category, with the lowest sensitivity reaching 94.7% and a precision of 95.9%. The overall accuracy is 97.7%, and the misclassification rate is 2.3%. Furthermore, Figure 3d describes the AUC for each class, which is nearly one for all categories. 

### 4.5. PCA

The principal component analysis is employed to select the most significant features that represent the four classes. Depending on a 95% variance among features, the most independent features are selected. As shown in the corresponding Figure 5, three principal components describe most of the variability in the data. However, the rest features have low significance in class representation.

Figure 6 shows the relationship between two principal components. The scatter representation visually shows how these two principals are capable of discriminating between classes. The clustering describes the classification capability of these two PCAs, where the red cluster indicates HSIL, the dark green cluster indicates normal, the cyan color represents LSIL, and the purple grouping distinguishes the SCC. The three significant features are exploited to train polynomial SVM. The corresponding confusion matrix in Figure 2e shows the performance of the classifier using the three independent features. The capability of the SVM to discriminate HSIL is low in terms of sensitivity and precision. On the other hand, recall and precision are low for LSIL. The normal class is the best distinguished, with a sensitivity and PPV of 100%. The precision of the SCC class is lower, at 87.2%, whereas the sensitivity is a moderate value that does not exceed 90%. The overall accuracy of the designed SVM using the most significant features is 90%, and the misclassification rate reaches 10%, which is too high. Moreover, Figure 3e illustrates the AUC for each class, which is the lowest in the SCC class with 0.95, and the highest in the normal class where the AUC is one. 

### 4.6. ICA

Forty features are passed to the independent component analysis algorithm to achieve the best independent and representative features among all. The best six features are a candidate. Figure 7 illustrates the scatter representation between the best two independent components. The grouping of the scattered points indicates the capability of the ICA to select the best representative features. According to Figure 6, the red group represents the HSIL class, the dark green cluster illustrates the normal (negative) class, the cyan bunch shows the LSIL class, and the purple color describes the SCC category. The best six independent features are passed to the third polynomial SVM. The corresponding confusion matrix shown in Figure 2f illustrates the output of the test phase. The best results were obtained in the LSIL class, with the sensitivity and precision reaching 100%. However, the lowest recall values in both the HSIL and the SCC classes were 97.3%. Furthermore, the lowest precision value in the SCC was 96.1%. On the other hand, the precision value of the LSIL was 100%. The overall accuracy using the ICA and SVM is 98.3% for all four classes, with a misclassification rate of 1.7%. Finally, Figure 3f shows the AUC for all the classes that are almost equal to one.

Figure 8 shows the comparison between the features engineering algorithm and its impact on the accuracy of the SVM classifier in discriminating whole-slice cervical images. The same data are shown in tabular form in Table 2. As illustrated in Figure 7, the highest accuracy achieved was by the CCA feature fusion, with a maximum accuracy reaching 99.7%. However, the performance of the other algorithms is almost the same with slight differences, apart from PCA, which exhibits the lowest accuracy value. These results show the influence of various feature processing algorithms on obtaining accurate computer-aided diagnosis systems. 

Table 3 shows the study comparison for the most recent studies that used data fusion on cervical cancer images. All mentioned studies showed the effectiveness of data fusion in improving the classification accuracy of cervical cancer. Comparing the previous studies that focused on cervical cancer diagnosis, the proposed approach in this paper achieves the highest accuracy with automated features. This paper deals with whole-slice cervical images, ignoring the overlapping and non-overlapping issues for cells. On top of that, all the previous studies focused on the diagnostics of single cells, whereas this paper deals with the whole-slice image, which is more practical for physicians and medical fields. Due to the limited work on feature-level fusion in cervical cancer, other studies with different medical diagnostic problems were shown in Table 4. These studies were selected based on the feature fusion technique used. All of the studies in Table 4 used data fusion analysis on the feature level. All the studies showed an improvement in classification accuracy when using feature-level fusion or selection. In our paper, we have adopted some of these existing methods. The studies listed in Table 3 and Table 4 have different perspectives on dealing with data fusion. They could be grouped into two perspectives: The first perspective is the data level that is being fused (feature level, matching score, or decision-level fusion) listed in Table 3. The second perspective is on the method used for fusion, the approaches mainly used either feature reduction techniques (such as PCA, ICA, and LASSO), or feature fusion techniques (such as CCA and DCA) listed in Table 4. These approaches have been used to fuse different types of data to enhance diagnostic decisions.

### 4.7. Computational Complexity

As explained above, extracting the features using DL models has demanded substantial time, which took hours of computation. Thereafter, feature fusion and SVM analyses have required seconds of computational time for each of the considered techniques. Therefore, the considered six techniques have demanded relatively comparable computational complexity when they are run on the same machine. 

### 4.8. Future Work and Real-Life Applications

To the best of our knowledge, this paper presents a unique approach of using ten pre-trained DL models with the most common feature selection techniques to diagnose whole-slice cervical images. The relatively high level of accuracy obtained herein can act as a background to building robust and reliable computer-aided detection and diagnosis systems for assessing colposcopy images. These findings can help reduce the mortality rate and enhance the chances of survival among women. Further enhancement on the proposed approach of analysis can be implemented in future works to expand the extracted features and to provide more robust results for medical diagnosis under different deep learning models.

## 5. Conclusions

This paper has focused on employing feature fusion techniques to enhance the classification accuracy of cervical cancer. It involved the generation of a new, uncorrelated dataset of features while faithfully conveying the output information. Using the new dataset of features, we have been able to reduce the dimension of feature space without degrading the performance of disease classification. This paper constructed a comparative analysis of the existing feature fusion techniques to extract the best representative features from ten independent datasets. These datasets came from ten pre-trained DL models, which were trained on a huge ImageNet database. Our approach to this analysis involved applying six sequential steps. The first step consisted of collecting a cytology dataset that contained 1000 samples for four different cervical cancer classes. The second step performed image augmentation, which was then followed by extracting features using CNN from ten DL models (4 features for each DL model for a total of 40 features). The next step concatenated all features from the ten DL models to be fed into the feature fusion step. Step five applied six different feature fusion techniques to extract features. Finally, the extracted features were input into an SVM to test the classification performance. The approach of this analysis revealed the highest accuracy of 99.7% using CCA fusion. The key benefit was reducing the number of features introduced to SVM and obtaining state-of-the-art accuracy. Therefore, the use of data fusion at the feature level, which was proposed in this paper, can indeed enhance classification accuracy for colposcopy images. The presented approach herein can be used as a guideline for other CAD medical applications to aid diagnostic decisions.

## Figures and Tables

**Figure 1 bioengineering-10-00105-f001:**
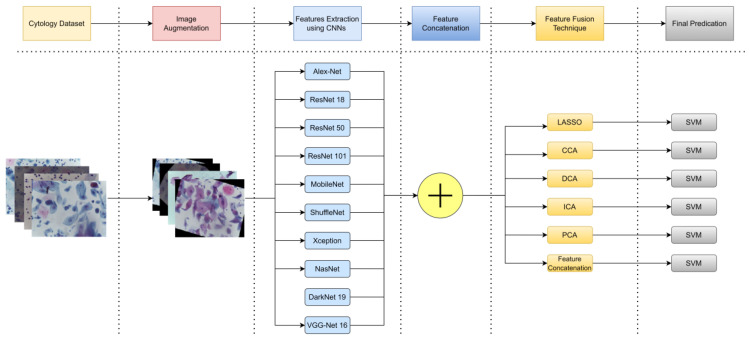
The proposed method. Showing all sequential steps of the proposed methodology in this paper.

**Figure 2 bioengineering-10-00105-f002:**
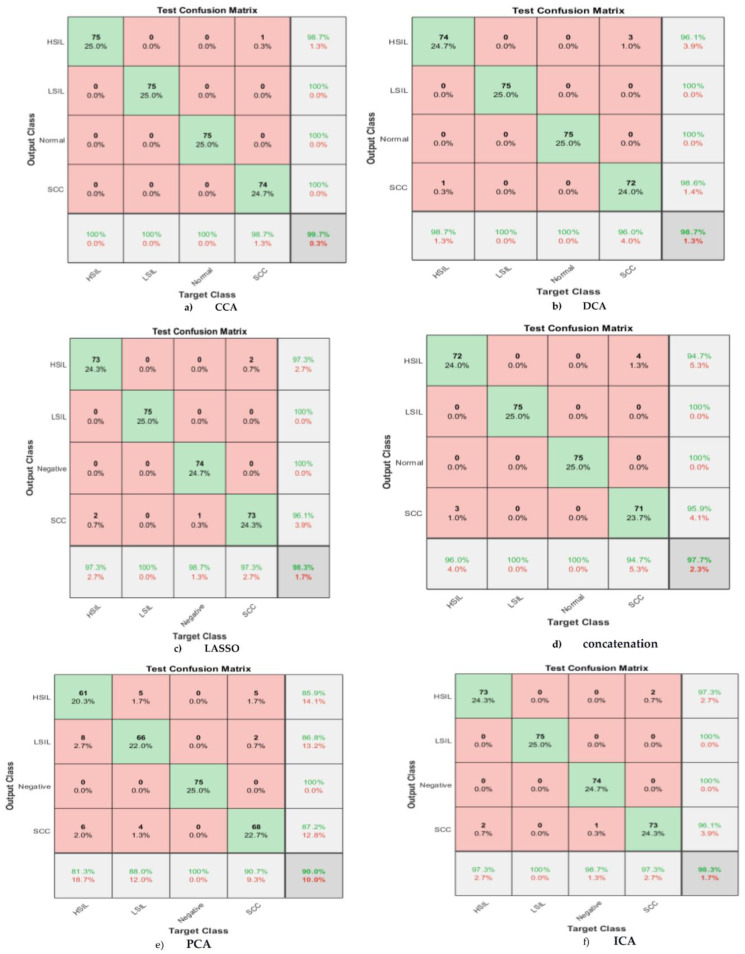
Six confusion matrices of the SVM model when considering six feature fusion techniques. (**a**) Using CCA, (**b**) DCA, (**c**) LASSO, (**d**) concatenation, (**e**) PCA, and (**f**) ICA. The matrices show the performance of the SVM after using different fusion techniques.

**Figure 3 bioengineering-10-00105-f003:**
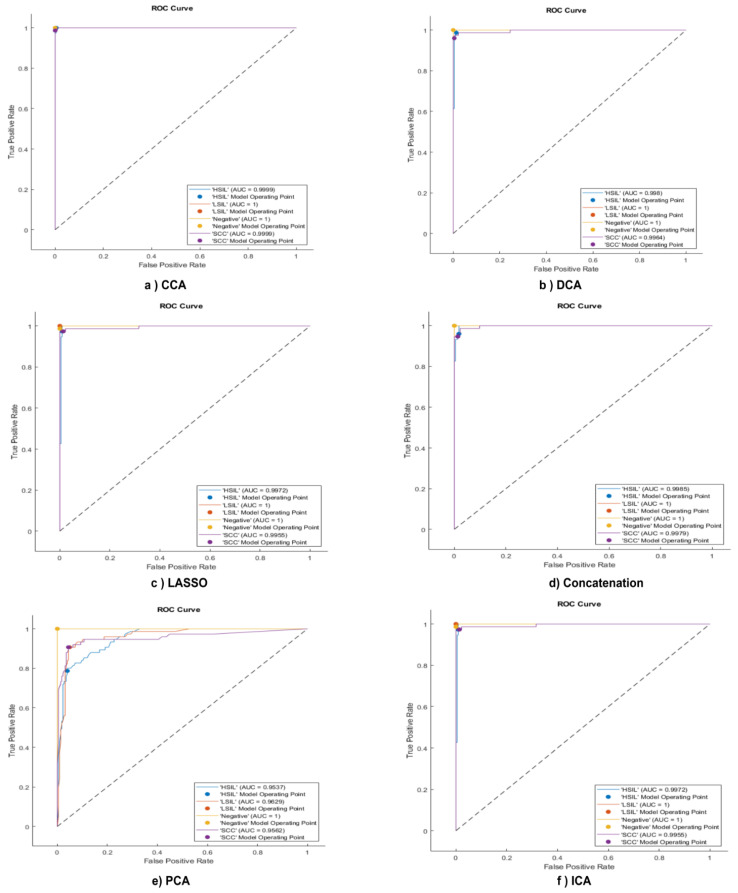
Six receiver operating characteristic curves of the SVM model when considering six feature fusion techniques. (**a**) Using CCA, (**b**) DCA, (**c**) LASSO, (**d**) concatenation, (**e**) PCA, and (**f**) ICA. The figure shows the performance of the SVM via AUC after using different fusion techniques.

**Figure 4 bioengineering-10-00105-f004:**
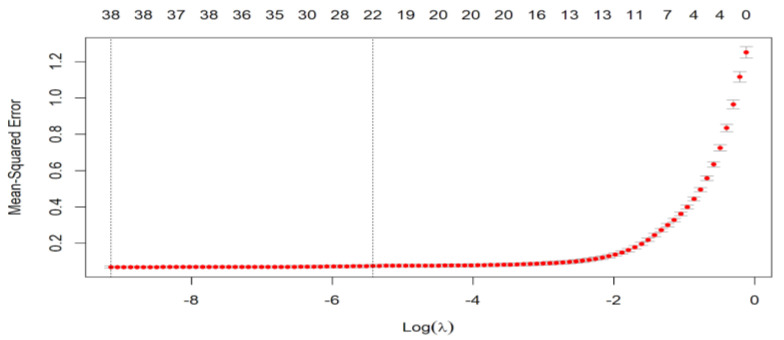
MSE of LASSO model. Showing how the number of features selected is affected by the MSE value.

**Figure 5 bioengineering-10-00105-f005:**
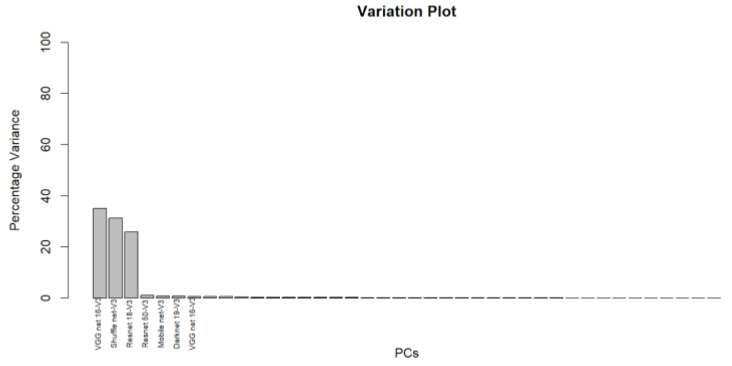
Percentage variance of each feature according to PCA. The first three features contribute the most to the variability of the data.

**Figure 6 bioengineering-10-00105-f006:**
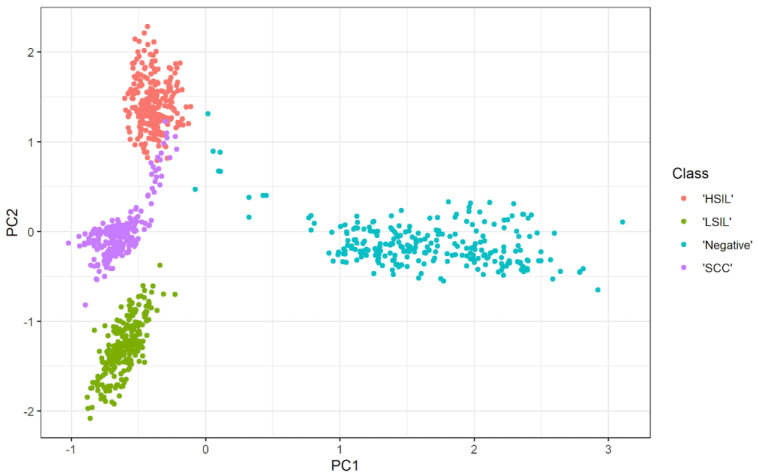
Scatter plot showing the first two principal components and how they visually discriminate between the four classes. The figure shows the effectiveness of separation between classes after selecting the most representative components using PCA.

**Figure 7 bioengineering-10-00105-f007:**
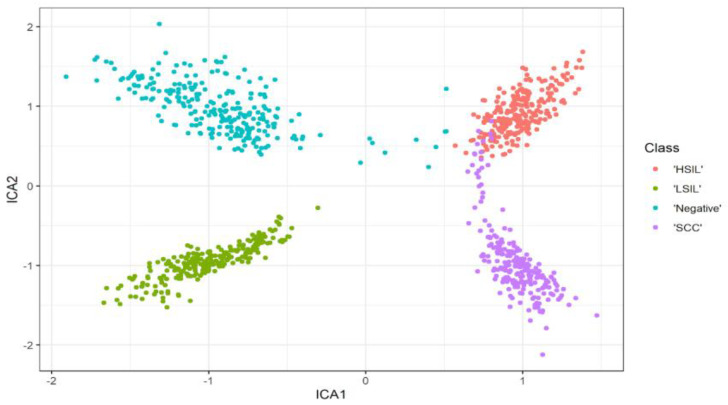
Scatter plot showing the first two independent components and how they discriminant between the four classes. The figure shows the effectiveness of separation between classes after selecting the most representative components using ICA.

**Figure 8 bioengineering-10-00105-f008:**
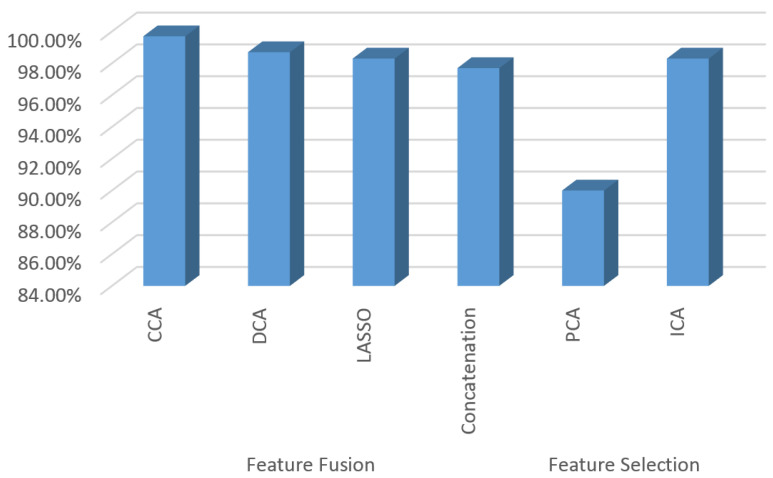
Bar plot showing the accuracy of the SVM when using different feature fusion techniques.

**Table 1 bioengineering-10-00105-t001:** The number of images before and after augmentation for abnormal cells. After augmentation the number of images becomes equal.

Abnormal Cells	Before Augmentation	After Augmentation
1. Low-grade squamous intraepithelial lesion (LSIL)	113	250
2. High-grade squamous intraepithelial lesion (HSIL)	163	250
3. Squamous cell carcinoma (SCC)	74	250

**Table 2 bioengineering-10-00105-t002:** Comparison between various scenarios. Showing the SVM accuracy for the six different feature analysis techniques.

Data Fusion Method	Number of Features Selected or Fused from the Original 40	SVM Accuracy
Concatenation	40	97.7%
LASSO	19	98.3
DCA	6	97.3%
CCA	6	99.7%
PCA	3	90%
ICA	3	98.3%

**Table 3 bioengineering-10-00105-t003:** Comparison with literature study used on cervical cancer images. The mentioned study focuses on using a fusion technique for the cervical cancer classification problem.

Study Author (Year)	Feature Fusion Method	Number of Fused Datasets	Best Accuracy
Alquran et al. (2022)	CCA	Two datasets from Shuffle Net and novel Cervical Net	99.1 (four-class classifications)
Liu et al. (2022)	Multilayer perceptron module	Two (CRIC and SIPaKMeD)	91.7 (eleven-class classifications)
Rahman et al. (21)	Late fusion	SIPaKMeD dataset	99.14 % (five-class classification)
Hussain et al. (2020)	Ensemble classifier based on selecting the best three DL models	Six datasets from (Alexnet, Vgg-16, Vgg-19, Resnet-50, Resnet-101, and Googlenet)	97% (two classes)
This paper	LASSO, CCA, DCA, PCA, and ICA	Ten datasets from (Alex Net, Resnet 18, 50, and 10, Mobilenet, Shufflenet, Xception, Nasnet, Darknet 19, and VGG Net 16)	99.7% (four classes)

**Table 4 bioengineering-10-00105-t004:** Most recent studies show the effectiveness of data feature fusion in improving classification accuracy on other medical diagnostic problems. Most of the mentioned feature fusion methods were selected as a part of our comparative study.

Study Author (Year)	Classification Problem	Feature Fusion Method
Fan et al. (2019)	Breast cancer prediction	CCA
Zhang et al. (2021)	Pancreatic ductal adenocarcinoma prediction	PCA, LASSO, Boruto, and proposed feature fusion method by Zhang et al.
Wnag et al. (2021)	COVID-19 classification	DCA
Haghighat et al. (2016)	Multimodal biometric recognition	DCA
This paper	Cervix cancer images four classes	LASSO, CCA, DCA, PCA, and ICA

## Data Availability

The dataset in this study was obtained from the Liquid-based cytology pap smear images for multi-class diagnosis of cervical cancer, which was published in Mendeley on 15 November 2019. It consists of 963 images that were collected from 460 patients and that were diagnosed into four classes. This dataset has been publicly available online since 2019. It is available on the corresponding website: https://data.mendeley.com/datasets/zddtpgzv63/3 (accessed on 15 March 2022).

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
