# Peer review of "Deep Feature Engineering in Colposcopy Image Recognition: A Comparative Study"

_bioengineering, 2023, doi:10.3390/bioengineering10010105_

Round 1

Reviewer 1 Report

I will list some of my major concerns in the following, which might help the authors to rebuild their current approach.
1. Literature review done by the authors is not comprehensive. I suggest authors to perform an exhaustive literature review. Authors should clearly demonstrate why their proposed approach is better than existing approaches in the literature.  I did not find some of the state-of-the-art descriptions related to previously published related works including more recently published works related in the literature review.  In this regard, I advise good papers, the author may see it for the improvement of the technical strength. the papers are:). Personal Mobility in Metaverse With Autonomous Vehicles Using Q-Rung Orthopair Fuzzy Sets Based OPA-RAFSI Model, Visual saliency guided complex image retrieval, A novel CNN based security guaranteed image watermarking generation scenario for smart city applications, Impact of digital fingerprint image quality on the fingerprint recognition accuracy, Four-image encryption scheme based on quaternion Fresnel transform, chaos and computer generated hologram, A survey of deep active learning.
2. The manuscript shows some shortcomings with respect to the logical flow of ideas. In particular, I miss a clear link between the research gap and the answering approach thereafter.
3. It is not very clear which components of your approach are a novelty and which of them are taken from previous studies.
4.
Explain the complexity and overhead analysis of the proposed approach.

5. In the Conclusions Section include a short comment on potential applications to benefit from the proposed methodology

Author Response

Reviewer 1

I will list some of my major concerns in the following, which might help the authors to rebuild their current approach.

  1. Literature review done by the authors is not comprehensive. I suggest authors to perform an exhaustive literature review. Authors should clearly demonstrate why their proposed approach is better than existing approaches in the literature.  I did not find some of the state-of-the-art descriptions related to previously published related works including more recently published works related in the literature review.  In this regard, I advise good papers, the author may see it for the improvement of the technical strength. the papers are:). Personal Mobility in Metaverse With Autonomous Vehicles Using Q-Rung Orthopair Fuzzy Sets Based OPA-RAFSI Model, Visual saliency guided complex image retrieval, A novel CNN based security guaranteed image watermarking generation scenario for smart city applications, Impact of digital fingerprint image quality on the fingerprint recognition accuracy, Four-image encryption scheme based on quaternion Fresnel transform, chaos and computer generated hologram, A survey of deep active learning.

Thank you for your comment. The literature review has been updated to clearly demonstrate the proposed approach in this paper. Unfortunately, the suggested references by the reviewer are not closely related to the scope of this paper. Please refer the update literature review in section 2 in the revised manuscript.  Mainly we have focused on papers that used feature engineering methods on cervical cancer classification problems. Due to limited studies in this area, we have chosen some recent papers that used specifically feature level fusion on other medical diagnostic problems. After that, we have adopted those methods that improved classification accuracy and applied them on cervical cancer 4-class classification.

  1. The manuscript shows some shortcomings with respect to the logical flow of ideas. In particular, I miss a clear link between the research gap and the answering approach thereafter.

Thank you for your comment. We have updated the abstract, introduction, related work, discussion, and conclusion to address the shortcomings. Please refere tot the updated secions.

  1. It is not very clear which components of your approach are a novelty and which of them are taken from previous studies.

Thank you for your comment. We have updated the abstract, introduction, related work, discussion, and conclusion to clarify the novelty and which part was taken from previous studies. Please refer to the updated sections

  1. Explain the complexity and overhead analysis of the proposed approach.

Thank you for your comment, we have added a subsection to the discussion titled ‘computational complexity’. Please refer to the section for more details.

It has been found that the considered six methods demand relatively comparable computational complexity when they are run on the same machine

  1. In the Conclusions Section include a short comment on potential applications to benefit from the proposed methodology

Thank you for your comment. In the first paper version the conclusion was embedded in the last paragraph of the discussion. However, in the second version of the paper, we have added a separate conclusion section. And expanded the discussion part, where there is a subsection titled ‘future work and real life applications’.

Reviewer 2 Report

The manuscript sounds technically poor; however, I have following concerns should be addressed before any decision.  

1.      Please explain in your captions of figure and title of table, why are these tables or figures necessary in your paper? What are the purposes and what are the message you want to deliver via these figures and tables?

2.      The current metrics might not be sufficient to judge the performance of the model holistically. Please enhance the result analysis part of your paper.

3.      The existing literature should be classified and systematically reviewed, instead of being independently introduced one-by-one.

4.       In the introduction section, the motivations of the proposed access control model must be included in detail. The section numbering must be changed in the paper organization paragraph.

5.      The abstract is too general and not prepared objectively. It should briefly highlight the paper's novelty as what is the main problem, how has it been resolved and where the novelty lies?

6.      The 'conclusions' are a key component of the paper. It should complement the 'abstract' and normally used by experts to value the paper's engineering content. In general, it should sum up the most important outcomes of the paper. It should simply provide critical facts and figures achieved in this paper for supporting the claims.

7.      For better readability, the authors may expand the abbreviations at every first occurrence.

8.      The author should provide only relevant information related to this paper and reserve more space for the proposed framework.

9.      The theoretical perceptive of all the models used for comparison must be included in the literature.

10.   What are the real-life use cases of the proposed model? The authors can add a theoretical discussion on the real-life usage of the proposed model.

11.   The related works section is very short and no benefits from it. I suggest increasing the number of studies and add a new discussion there to show the advantage.  

12.   The descriptions given in this proposed scheme are not sufficient that this manuscript only adopted a variety of existing methods to complete the experiment where there are no strong hypothesis and methodical theoretical arguments. Therefore, the reviewer considers that this paper needs more works.

13.   Key contribution and novelty has not been detailed in manuscript. Please include it in the introduction section

Author Response

Reviewer 2

The manuscript sounds technically poor; however, I have following concerns should be addressed before any decision.  

  1. Please explain in your captions of figure and title of table, why are these tables or figures necessary in your paper? What are the purposes and what are the message you want to deliver via these figures and tables?

Thank you for your comment. Captions of figures and table shave been revised. Please see the updated revisions.

  1. The current metrics might not be sufficient to judge the performance of the model holistically. Please enhance the result analysis part of your paper.

Thank you for your comment. AUC analyses for each feature fusion technique have been added as figure 3.

  1. The existing literature should be classified and systematically reviewed, instead of being independently introduced one-by-one.

Thank you for your comment. The related study section has been revised. Please refer to the updated section.

  1. In the introduction section, the motivations of the proposed access control model must be included in detail. The section numbering must be changed in the paper organization paragraph.

Thank you for your comment. We have revised the introduction and added details about motivation and the aim of the paper. Please refer to the updated introduction.  We have corrected the section numbering. We are sorry about this typo mistake.

  1. The abstract is too general and not prepared objectively. It should briefly highlight the paper's novelty as what is the main problem, how has it been resolved and where the novelty lies?

Thank you for your comment. The abstract has been carefully revised and more details have been added to objectively highlight the aim and main findings in the paper.

  1. The 'conclusions' are a key component of the paper. It should complement the 'abstract' and normally used by experts to value the paper's engineering content. In general, it should sum up the most important outcomes of the paper. It should simply provide critical facts and figures achieved in this paper for supporting the claims.

Thank you for your comment. In the first paper version, the conclusion was embedded in the last paragraph of the discussion. However, in the second version of the paper, we have added a separate conclusion section. Please refer to the new conclusion section.

  1. For better readability, the authors may expand the abbreviations at every first occurrence.

Thank you for your comment.  All abbreviations have been revised.

  1. The author should provide only relevant information related to this paper and reserve more space for the proposed framework.

Thank you for your comment. We have carefully updated the abstract, introduction, related work, discussion, and conclusion. Please refer to the updated sections.

  1. The theoretical perceptive of all the models used for comparison must be included in the literature.

Thank you for your comment.  All the applied methods in the paper have been referenced in the introduction, related work, and material and methods sections. Please refer to the sections for more details.

  1. What are the real-life use cases of the proposed model? The authors can add a theoretical discussion on the real-life usage of the proposed model.

Thank you for your comment. A subsection titled ‘future work and real life applications’ have been added to the discussion. Please refer to the updated discussion section.

  1. The related works section is very short and no benefits from it. I suggest increasing the number of studies and add a new discussion there to show the advantage.  

Thank you for your comment. The introduction and related work section are updated. More explanations have been added to the discussion. Please refer to the updated abstract, introduction, related work, and discussion sections.

  1. The descriptions given in this proposed scheme are not sufficient that this manuscript only adopted a variety of existing methods to complete the experiment where there are no strong hypothesis and methodical theoretical arguments. Therefore, the reviewer considers that this paper needs more works.

Thank you for your comment. We have conducted a comparison study to measure the effectiveness of different feature fusion techniques on cervical cancer classification. The feature dataset was extracted from ten well-known DL structures. All six data fusion techniques are existing in the literature. Moreover, the literature showed evidence that feature fusion showed improvement in classification accuracy (which is explained in the related study section in our paper and summarized in Tables 3 and 4). We applied these existing methods on cervical cancer classification problem. Please refer to the updated abstract, introduction, related work, and conclusion sections for more details.

  1. Key contribution and novelty has not been detailed in manuscript. Please include it in the introduction section

Thank you for your comment. The contribution/novelty have been clarified in the abstract, introduction and discussion. Please refer to the revised abstract, introduction, and conclusion for more details.

Reviewer 3 Report

In this paper, the authors have compared the effectiveness of six feature fusion techniques namely; Canonical Correlation Analysis (CCA), Discriminant Correlation Analysis (DCA), Least Absolute Shrinkage and Selection Operator (LASSO), Independent Com-ponent Analysis (ICA), Principal Component Analysis (PCA), and Concatenation. In general, this paper is well written and easy to follow. I would like to accept this paper if my following concerns are carefully addressed.

(1) The authors need to emphasise their contributions/novelties in the revision. In the current version, the authors did not discuss their contributions in detail.
(2) The proposed algorithm still can be improved if the ideas in the following papers are explored, i.e., "Making Sense of Spatio-Temporal Preserving Representations for EEG-Based Human Intention Recognition", "An Adaptive Semisupervised Feature Analysis for Video Semantic Recognition", and "A Semisupervised Recurrent Convolutional Attention Model for Human Activity Recognition". The authors are encouraged to discuss them in the revision.
(3) The authors should carefully proofread this paper and correct all the typos in the revision. In the current version, there are still some typos/grammar errors.
(4) Could the authors report the running time of the proposed algorithm? In this way, we can justify whether this algorithm can be applied to large-scale dataset.
Based on the above comments, I would like to accept this paper with minor revision.

Author Response

Reviewer 3

In this paper, the authors have compared the effectiveness of six feature fusion techniques namely; Canonical Correlation Analysis (CCA), Discriminant Correlation Analysis (DCA), Least Absolute Shrinkage and Selection Operator (LASSO), Independent Com-ponent Analysis (ICA), Principal Component Analysis (PCA), and Concatenation. In general, this paper is well written and easy to follow. I would like to accept this paper if my following concerns are carefully addressed.

1.The authors need to emphasise their contributions/novelties in the revision. In the current version, the authors did not discuss their contributions in detail.

Thank you for your comment. The contribution/novelty have been clarified in the abstract, introduction and discussion. Please refer to the revised abstract, introduction, and conclusion for more details.

  1. The proposed algorithm still can be improved if the ideas in the following papers are explored, i.e., "Making Sense of Spatio-Temporal Preserving Representations for EEG-Based Human Intention Recognition", "An Adaptive Semisupervised Feature Analysis for Video Semantic Recognition", and "A Semisupervised Recurrent Convolutional Attention Model for Human Activity Recognition". The authors are encouraged to discuss them in the revision.

Thank you for your comment. Unfortunately, the suggested references by the reviewer are not closely related to the scope of this paper.

  1. The authors should carefully proofread this paper and correct all the typos in the revision. In the current version, there are still some typos/grammar errors.

Thank for your comment. The whole paper has been carefully proofread.

  1. Could the authors report the running time of the proposed algorithm? In this way, we can justify whether this algorithm can be applied to large-scale dataset.

Thank you for your comment. Please refer to the added section titled ‘computational complexity’ in the discussion which answers your comment on computational complexity.

Reviewer 4 Report

The current study focused on feature fusion techniques in order to improve cervical cancer classification. Six different features engineering algorithms were compared and an accuracy of  99.7% was obtained for 4-class classification of Cervical cancer when using CCA as a feature fusion method. The presented study might pose potential clinical significance for cervical cancer diagnosis. There are several minor suggestions listed below.

1. The organization of the manuscript might be improved further. E.g. Could 2. Review of Study section be simplified and incorporated into the introduction section?

2. Line 378 should be numbered as 3.5? if so, the subsequent sequence numbers need to be changed.

3.The current study is retrospectively and further prospective validation or trial is expected in future.  

Author Response

Reviewer 4

The current study focused on feature fusion techniques to improve cervical cancer classification. Six different features engineering algorithms were compared and an accuracy of  99.7% was obtained for the 4-class classification of Cervical cancer when using CCA as a feature fusion method. The presented study might pose potential clinical significance for cervical cancer diagnosis. There are several minor suggestions listed below.

  1. The organization of the manuscript might be improved further. E.g. Could the “2. Review of Study” section be simplified and incorporated into the introduction section?

Thank you for your comment. The ‘introduction’ and ‘related work’ sections are updated. Moreover, the ‘review of study’ section has been revised and renamed as ‘related work’.

  1. Line 378 should be numbered as 3.5. if so, the subsequent sequence numbers need to be changed.

Thank you for your comment. The section 3.4 numbering has been changed to 3.5 and the subsequent total have been changed.

3. The current study is retrospectively and further prospective validation or trial is expected in future.  

Thank you for your comment. Explanation have been added in the discussion showing novelty of our approach compared with other studies, future work, the real-life usage of the proposed model. And a separate conclusion section is now present.

Round 2

Reviewer 2 Report

The paper is relatively improved to the previous version. It can be considered for publication.